

# Which patients are more likely to experience compensatory hyperhidrosis after endoscopic thoracic sympathectomy: a meta-analysis and systematic review

Zhi-yi Lin and  Min Lin

Department of Thoracic Surgery, First Affiliated Hospital of Fujian Medical University, Fuzhou, Fujian, China

## ABSTRACT

**Background**. Compensatory hyperhidrosis (CH) is a common consequence of sympathectomy, which can adversely affect patients' quality of life after surgery. Understanding the factors that influence the occurrence of CH and severe compensatory hyperhidrosis (SCH) is crucial for effective management and counseling of patients undergoing this procedure.

**Materials and Methods**. We registered the protocol in International Prospective Register of Systematic Reviews (CRD42024592389) and following PRISMA guidelines. We searched PubMed, EMBASE, and Web of Science databases for studies published up to September 11, 2024. A systematic literature search identified a total of 10 studies involving 3,117 patients. The primary outcome was the number of CH or SCH. The secondary outcome was the weighted mean difference calculated based on identified related factors. When pooling results or conducting a meta-analysis was not feasible, the study findings were presented in a narrative descriptive format.

**Results**. The overall incidence of CH was found to be 0.62 (95% confidence interval CI [0.51–0.72]), and four studies totaling 1,618 patients regarding the occurrence of severe compensatory hyperhidrosis, the overall incidence of CH was found to be 0.23 (95% CI [0.12–0.34]). Older age, higher body mass index (BMI) and smoking history correlated positively with CH incidence. In addition, higher BMI level is also associated with the occurrence of SCH (1.20 95% CI [1.01–1.39], $p < 0.0001$).

**Conclusion**. The findings of this meta-analysis highlight important demographic and lifestyle factors that contribute to the development of CH and SCH following sympathectomy. Older patients, smokers, and those with higher BMI may be at greater risk for these conditions.

Corresponding author
Min Lin, 1068919@qq.com

## INTRODUCTION

Hyperhidrosis is primarily characterized by excessive sweating under normal circumstances, resulting in noticeable perspiration in areas such as the hands, armpits, soles of the feet, and even the face, either continuously or intermittently, which severely affects normal daily activities (*Augustin et al., 2013*). The prevalence of hyperhidrosis varies globally, with

reported rates ranging from 1% to 16.7%, depending on studies from different countries and regions. These variations may be influenced by genetic factors, climate conditions, and lifestyle choices (*Ricchetti-Masterson et al., 2018*; *Stefaniak et al., 2013*).

Endoscopic sympathectomy (ETS) is performed through small incisions in the patient's chest using thoracoscopy. Based on the type of hyperhidrosis and the surgeon's experience, electrical cautery is used to sever the sympathetic nerve trunk associated with hyperhidrosis, effectively reducing or halting abnormal gland stimulation, making it the preferred treatment with the most evident outcomes (*Liu et al., 2022*). Nonetheless, one of the most frustrating postoperative side effects is compensatory hyperhidrosis (CH), which leads to sweating in different anatomical regions, most frequently the abdomen, chest, back, and groin. The incidence of CH varies widely, from 3% to 98% (*Bryant & Cerfolio, 2014*). CH not only causes physical discomfort, such as the need to frequently change clothes and avoid certain activities, but also leads to significant psychological distress, including anxiety and social withdrawal. Patients with severe compensatory hyperhidrosis (SCH) may experience more severe symptoms, such as persistent and excessive sweating, which can interfere with daily activities and work, leading to a significant decline in overall health and regret over their surgical choices.

Despite the prevalence of CH and SCH, there are still significant gaps in our understanding and management of these conditions. Predictive models for identifying patients at higher risk of developing CH and SCH are inadequate. Effective long-term management strategies for severe cases are also lacking, leaving many patients with limited options for symptom relief. Additionally, the psychological impact of CH and SCH on patients remains underexplored, with few studies comprehensively assessing the extent of anxiety, depression, and social dysfunction associated with these conditions. We aims to systematically explore the influencing factors of CH and SCH following ETS to enhance clinical practice and improve patient outcomes.

# MATERIALS AND METHODS

## Study design and protocol

This systematic review and meta-analysis were conducted following the Preferred Reporting Items for Systematic Reviews and Meta-Analyses guidelines (PRISMA). The study protocol was prospectively registered with the International Prospective Register of Systematic Reviews under the registration number CRD42024592389, ensuring transparency and minimizing the risk of bias in study selection and data extraction. The protocol outlined the objectives, inclusion and exclusion criteria, search strategy, data extraction methods, and statistical analysis plan.

## Literature search strategy

A comprehensive literature search was conducted in the PubMed, EMBASE, and Web of Science databases, covering studies published from their inception to September 11, 2024. The search strategy was developed in consultation with a medical librarian to maximize both sensitivity and specificity in identifying relevant studies. The primary search terms included: ("compensatory sweating"). To ensure a broad yet targeted search, additional

terms such as ("endoscopic thoracic sympathectomy" OR "hyperhidrosis surgery" OR "ETS") and ("hyperhidrosis" OR "excessive sweating") were incorporated. The search strategies for each database were as follows:

For PubMed, the search formula was: ("facial hyperhidrosis" OR "sweating disorder" OR "excessive sweating" OR "facial hyperhidrosis" OR "sweating disorder" OR "excessive sweating" OR "facial blushing" OR "flushing" OR "blushing" OR "flushing") AND ("endoscopic thoracic sympathectomy" OR "thoracic sympathectomy" OR "ETS" OR "sympathetic surgery" OR "surgical treatment"). For EMBASE, the search formula was: (('facial hyperhidrosis':ti,ab OR 'sweating disorder':ti,ab OR 'excessive sweating':ti,ab OR 'facial blushing':ti,ab OR 'flushing':ti,ab OR 'blushing':ti,ab) AND ('endoscopic thoracic sympathectomy':ti,ab OR 'thoracic sympathectomy':ti,ab OR 'ETS':ti,ab OR 'sympathetic surgery':ti,ab OR 'surgical treatment':ti,ab)). For Web of Science, the search formula was: ("facial hyperhidrosis" OR "sweating disorder" OR "excessive sweating" OR "facial hyperhidrosis" OR "sweating disorder" OR "excessive sweating" OR "facial blushing" OR "flushing" OR "blushing" OR "flushing") AND ("endoscopic thoracic sympathectomy" OR "thoracic sympathectomy" OR "ETS" OR "sympathetic surgery" OR "surgical treatment").

The references of relevant articles and systematic reviews were manually screened to identify any additional studies that met the inclusion criteria.

## Study selection and eligibility criteria

The study selection process followed the Preferred Reporting Items for Systematic Reviews and Meta-Analyses (PRISMA) guidelines. Two independent reviewers (Zhi-yi Lin and Hong Xiao) conducted a two-stage screening process: an initial review of titles and abstracts, followed by a full-text review of articles that met the preliminary inclusion criteria. Any disagreements between the reviewers were resolved through discussion, or by consulting a third reviewer (Min Lin) if consensus could not be reached.

Inclusion criteria: (a) studies involving participants diagnosed with hyperhidrosis who underwent ETS; (b) reported the number of patients with CH or SCH, evaluation of the relationship between any risk factors and CH or SCH; (c) study designs including cohort studies, case-control studies, cross-sectional surveys, or controlled trials.

Exclusion criteria: (a) non-human studies; (b) review articles, case reports, studies with fewer than 30 participants, and studies lacking complete data; (c) patients underwent endoscopic thoracic clipping or computed tomography guided percutaneous thoracic sympathetic radiofrequency thermocoagulation.

All citations were managed using EndNote X9 (Clarivate Analytics, Philadelphia, PA, USA) to facilitate the removal of duplicates and streamline the selection process. The selection process, including the number of studies identified, screened, excluded, and included, is detailed in a PRISMA flow diagram (Fig. 1).

## Data extraction

Two independent reviewers (Zhi-yi Lin and Hong Xiao) extracted data using a standardized form, which was pilot-tested on a subset of studies. The extracted information included

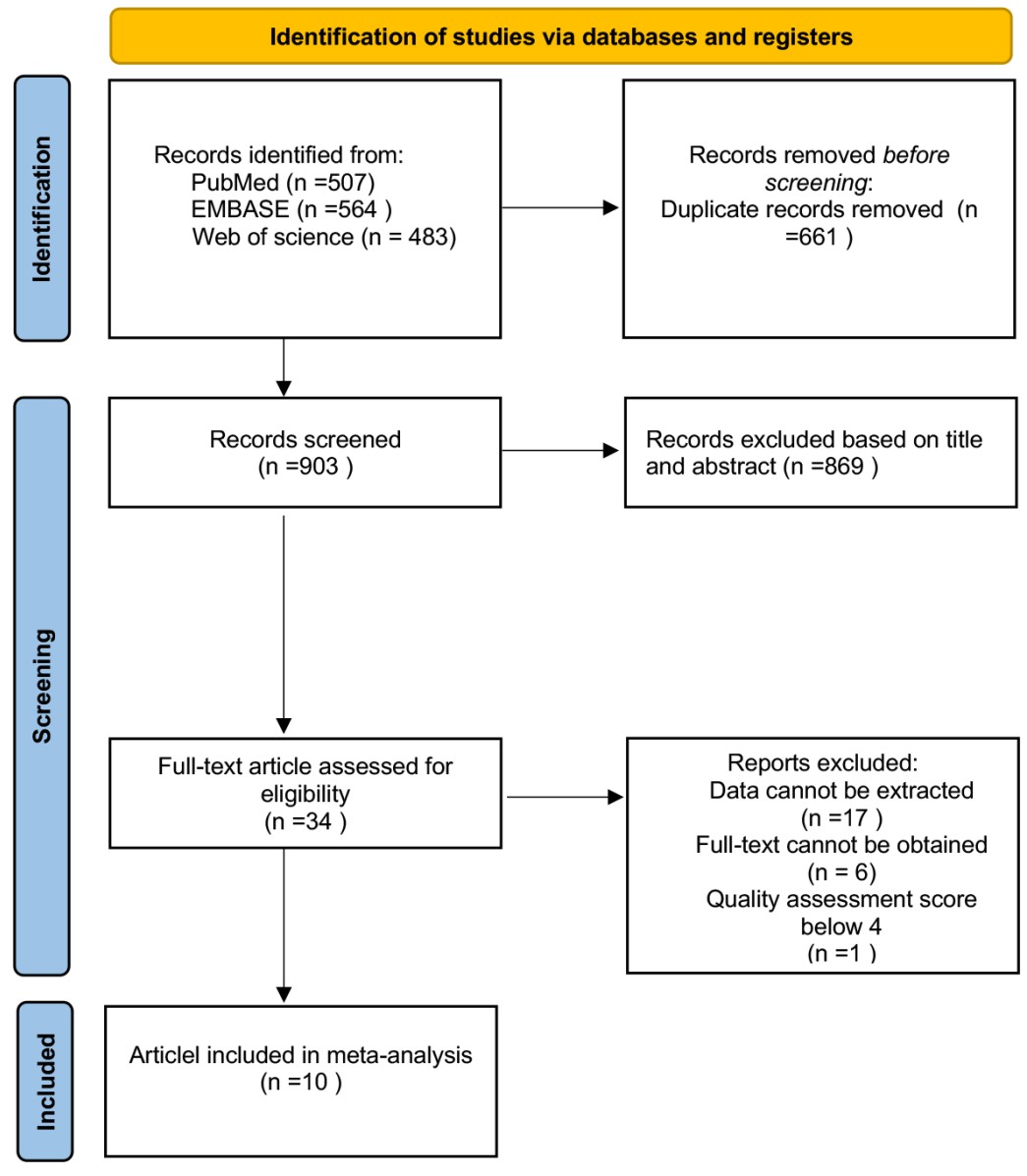

**Figure 1** Flow diagram of the included studies in this meta-analysis.

study characteristics (authors, year of publication, country, study design), the sample size, the number of patients with CH or SCH, any type of related risk factors (such as age, sex, smoking history, BMI), intervention characteristics (surgical technique, level of sympathectomy). Discrepancies were resolved through discussion, or with input from a third reviewer (Min Lin). The final dataset was cross-verified for accuracy and completeness. Detailed data extracted from these studies are presented in Tables 1 and 2.

Lin and Lin (2025), *PeerJ*, DOI 10.7717/peerj.19097

**Table 1** Characteristics of the records including factors are provided.

| Study | Year | Design | Nation | Surgical sites | N(n) | CH+ (n) | CHR% | Related factors | Definitions | Follow-up times | Quality |
|---|---|---|---|---|---|---|---|---|---|---|---|
| Masarwa et al. | 2024 | Retrospective | Palestine | R2 or R3 | 50 | 39 | 78 | Age, smoking history, and BMI | HDSS | 6 months | Moderate |
| Alkosha et al. | 2023 | Retrospective | Egypt | R3 or R3-R4 | 194 | 90 | 46 | Age, smoking history, BMI, and associated plantar hyperhidrosis | Subjective | 12 months | Moderate |
| Woo et al. | 2022 | Retrospective | Korea | R3 or R4 or R4-R5 | 231 | 199 | 86.1 | Age | HDSS | 20 months | Moderate |
| Carvalho et al. | 2022 | Retrospective | Portugal | R2-R4 | 41 | 13 | 31.7 | NA | Subjective | 6 months | Moderate |
| Kara et al. | 2019 | Prospective | Indian | R2-R4 or R2 or R3 or R4 | 74 | 45 | 60.8 | Age, smoking history, and BMI | Subjective | 12.6 ± 7.8 months | Moderate |
| Leiderman et al. | 2018 | Retrospective | Israel | R2-R4 or R2-R3 or R3-R4 or R2 or R3 or R4 | 1570 | 1076 | 68.5 | BMI | Subjective | 1 months | High |
| Rodríguez et al. | 2008 | Retrospective | Spain | R2-R4 or R2-R3 or R3-R4 | 406 | 223 | 54.9 | Sex | Subjective | 12 months | Moderate |

**Notes.**

CH+, compensatory hyperhidrosis positive; CHR, compensatory hyperhidrosis rate; R2, thoracic sympathectomy at the second thoracic level; R3, thoracic sympathectomy at the third thoracic level; R4, thoracic sympathectomy at the forth thoracic level; R5, thoracic sympathectomy at the fifth thoracic level; NA, not applicable; BMI, body mass index; HDSS, Hyperhidrosis disease severity scale.

*Masarwa et al., 2024*; *Alkosha et al., 2023*; *Woo et al., 2022*; *Carvalho et al., 2022*; *Kara et al., 2019*; *Leiderman et al., 2018*; *Rodríguez et al., 2008*.

Lin and Lin (2025), *PeerJ*, DOI 10.7717/peerj.19097

**Table 2** **Characteristics of the seven records including factors related to severe compensatory hyperhidrosis are provided.**

| Study | Year | Design | Nation | Surgical sites | N(n) | SCH+ (n) | SCHR% | Related factors | Definitions | Follow-up times | Quality |
|---|---|---|---|---|---|---|---|---|---|---|---|
| Adhami et al. | 2023 | Retrospective | Australia | R2 or R3 or R4 | 298 | 50 | 16.8 | Age, smoking history | Need to change clothes | 4.9 ± 1.8 years | Moderate |
| Toolabi et al. | 2022 | Retrospective | Iran | R2 or R3 or R4 or R3-R4 | 200 | 52 | 26 | BMI | Cannot stand | 10 years | Moderate |
| Moon et al. | 2020 | Prospective | Korea | R2 and R4-R7 | 53 | 26 | 49.05 | HRV | Need to change clothes | 3 months | Moderate |
| Leiderman et al. | 2018 | Retrospective | Israel | R2-R4 or R2-R3 or R3-R4 or R2 or R3 or R4 | 1570 | 84 | 7.9 | BMI | Need to change clothes | 1 month | High |

**Notes.**

CHR, compensatory hyperhidrosis rate; SCH+, severe compensatory hyperhidrosis positive; SCH-, severe compensatory hyperhidrosis negative; SCHR, severe compensatory hyperhidrosis rate; R2, thoracic sympathectomy at the second thoracic level; R3, thoracic sympathectomy at the third thoracic level; R4, thoracic sympathectomy at the forth thoracic level; R7, thoracic sympathectomy at the seventh thoracic level; NA, not applicable; BMI, body mass index; HRV, heart rate variability.

*Adhami & Bell, 2023*; *Toolabi et al., 2022*; *Moon et al., 2020*; *Leiderman et al., 2018*.

## Quality assessment

The quality of included studies was assessed independently by two reviewers using appropriate tools: For cohort and case-control studies, the Newcastle-Ottawa Scale (NOS) was applied. Quality assessments considered selection bias, exposure/outcome assessment, confounding, and statistical reporting. Disagreements were resolved through discussion or consultation with a third reviewer. Any study assessed as having a high risk of bias excluded from the final analysis. The quality of each study was taken into account when interpreting the results of the meta-analysis, and sensitivity analyses were conducted to test the robustness of conclusions based on the inclusion of lower-quality studies. The quality analysis results of the article are presented in Tables 1 and 2.

## Statistical analysis

The meta-analysis was conducted using Review Manager version 5.3 (Cochrane Collaboration, Oxford, UK) and StataSE 16.0 (StataCorp, College Station, TX, USA). A random-effects model was employed to calculate event rates and their corresponding 95% confidence intervals (CIs), in order to account for potential heterogeneity across studies. Statistical heterogeneity was assessed using the $I^2$ statistic and Cochran's Q test, with an $I^2$ value greater than 50% or a $P$-value less than 0.10 indicating significant heterogeneity. Additionally, subgroup analysis and meta-regression were performed to explore potential sources of heterogeneity, such as differences in patient demographics or surgical techniques. Publication bias was evaluated visually using funnel plots, and statistically using Egger's test. Sensitivity analysis was carried out by sequentially excluding individual studies to test the robustness of the overall results, ensuring that the findings were not overly influenced by any single study. The study selection process adhered to the PRISMA guidelines. A PRISMA flow diagram was provided to illustrate the details of study identification, screening, eligibility, and inclusion.

# RESULTS

## Eligible studies

In our initial literature search, we identified 1,554 articles. After removing duplicates, screening titles and abstracts, and reviewing full texts, a total of 10 articles met the inclusion criteria and were included in this meta-analysis (Masarwa et al., 2024; Adhami & Bell, 2023; Alkosha et al., 2023; Woo et al., 2022; Carvalho et al., 2022; Toolabi et al., 2022; Moon et al., 2020; Kara et al., 2019; Leiderman et al., 2018; Rodríguez et al., 2008). Among these 10 articles, six studies focused on the factors influencing the occurrence of compensatory sweating after sympathectomy, while three studies investigated factors influencing severe compensatory sweating rates post-sympathectomy. The remaining one article addressed both aspects (Tables 1 and 2). Figure 1 illustrates the PRISMA flowchart in the meta-analysis paper.

## Quality assessment and publication bias

Each article's quality was assessed using the Newcastle-Ottawa Scale, with a median quality score of six points falling within a range of six to seven points. Publication bias was evaluated

using Egger's test. Among the seven articles investigating CH occurrence rates, Egger's test results showed no evidence of publication bias ($P > 0.05$). However, for the four articles focusing on SCH occurrence rates, Egger's test indicated the presence of publication bias ($P < 0.05$). Even after applying the trim-and-fill method, the $P$-value remained below 0.05, indicating the robustness of the results, which were unaffected even with the addition of virtual studies.

## Occurrence of compensatory hyperhidrosis

A total of seven studies reported on 2,566 patients regarding the occurrence of compensatory hyperhidrosis. Due to high heterogeneity ($I^2 = 97\%$; Fig. 2), the random effect model was used for analysis, and the results showed that the incidence of CH was 0.62 (95% CI [0.51–0.72]). After sensitivity analysis, no sources of heterogeneity were found. The subgroup analysis showed that the incidence of CH in studies from Asia was higher than that in other continents (0.74 vs. 0.46, $P = 0.0007$; Fig. S1), which may be one source of heterogeneity. Subgroup analysis based on the definition of CH revealed that the incidence of CH defined using the HDSS was significantly lower than that defined by subjective assessment, suggesting a potential source of heterogeneity (Fig. S2). However, subgroup analysis with varying follow-up durations did not reveal any significant differences between the groups (Fig. S3). We further conducted a meta-regression based on whether the study originated from Asia, different definitions of CH, and varying follow-up times, yielding $p$-values of 0.055, 0.041, and 0.284, respectively. These results similarly indicate that differing definitions of CH may contribute to discrepancies in the reported incidence of CH. We compared the clinical parameters of patients with CH to those without CH to determine if there were any differences.

We compared the ages of patients with CH and those without CH; all 2,566 patients had reported ages. The random effects model showed that patients with CH were older than those without CH, with a WMD of 1.92 (95% CI [0.74–3.11], $p = 0.002$; Fig. 3A), indicating that older patients may be more likely to develop CH after undergoing ETS. This difference was particularly significant in non-Asian populations (Fig. S4). However, subgroup analyses based on varying definitions of CH (Fig. S5) and different follow-up durations (Fig. S6) did not reveal any significant age differences between the groups. A total of 939 male patients and 1,628 female patients underwent ETS surgery, but no correlation was found between gender and the occurrence of CH ($P = 0.82$; Fig. 3B). Five articles reported on a total of 764 patients regarding family history, with 144 patients having a family history and 620 without; due to low heterogeneity ($I^2 = 0\%$), a fixed effects model was used, and no differences were found in the occurrence of CH with or without a family history ($P = 0.45$; Fig. 3C). Sensitivity analysis for the above comparisons revealed no sources of heterogeneity.

Four studies investigated the smoking history of patients, with $I^2 = 89\%$ (Fig. S7). Sensitivity analysis indicated that one article (*Alkosha et al., 2023*) might be a source of heterogeneity, so after excluding this article, a meta-analysis was conducted on the remaining three studies, involving a total of 355 patients, with $I^2 = 70\%$. Among these, 57 patients had a smoking history, while 298 did not. The random effects model suggested

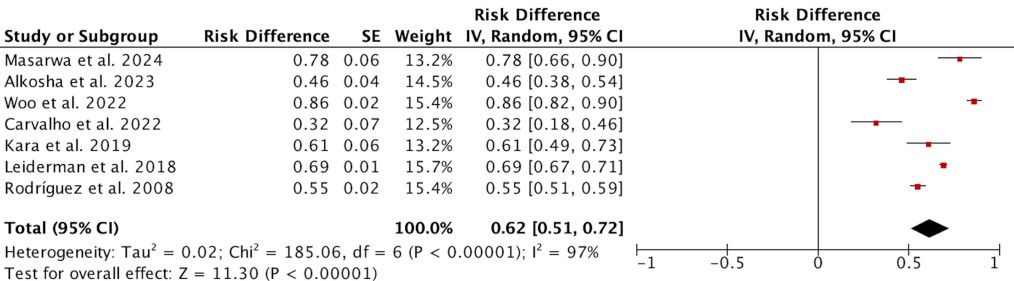

Figure 2 Forest plots of compensatory hyperhidrosis rate in patients after endoscopic thoracic sympathectomy. Note. *Masarwa et al., 2024*; *Alkosha et al., 2023*; *Woo et al., 2022*; *Carvalho et al., 2022*; *Kara et al., 2019*; *Leiderman et al., 2018*; *Rodríguez et al., 2008*.

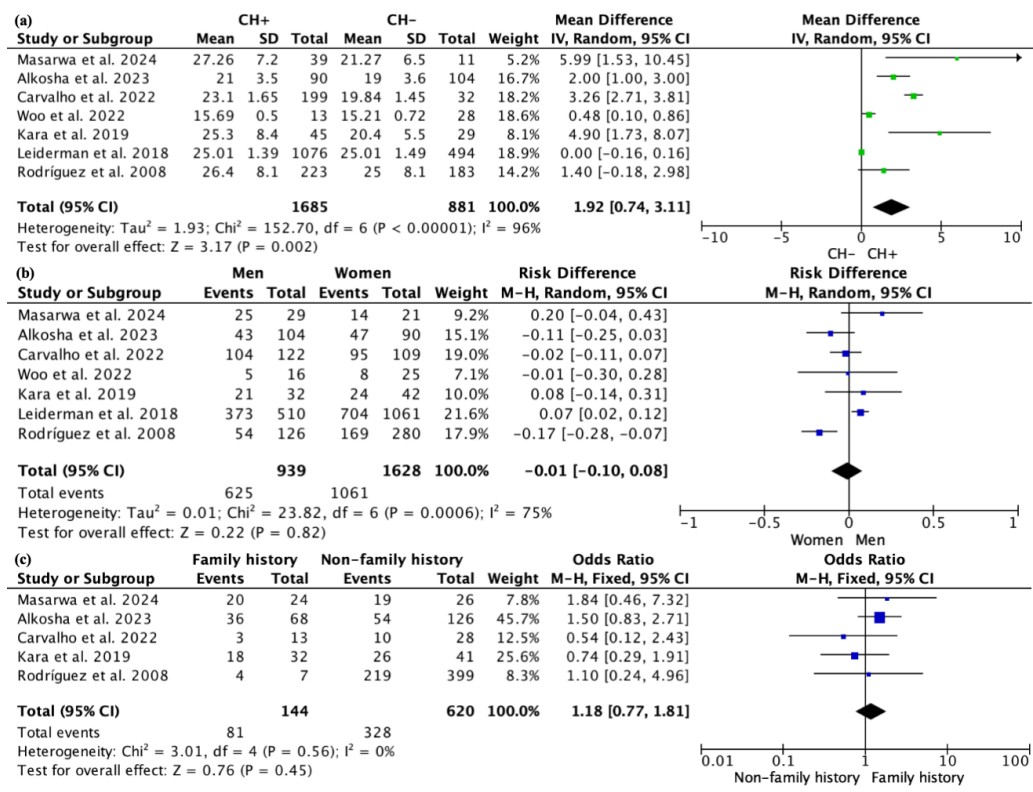

Figure 3 Forest plot comparing clinical parameters between patients with and without compensatory hyperhidrosis after endoscopic thoracic sympathectomy. Note. *Masarwa et al., 2024*; *Alkosha et al., 2023*; *Woo et al., 2022*; *Carvalho et al., 2022*; *Kara et al., 2019*; *Leiderman et al., 2018*; *Rodríguez et al., 2008*.

that the rate of CH occurrence after surgery was significantly higher in patients with a smoking history compared to those without ($P < 0.0001$; Fig. 3D).

Four studies reported BMI data for a total of 2,069 patients. Due to high heterogeneity ($I^2 = 97\%$), a random effects model was used. The results indicated that patients with a higher BMI were more likely to develop CH after ETS, with a WMD of 2.07 (95% CI

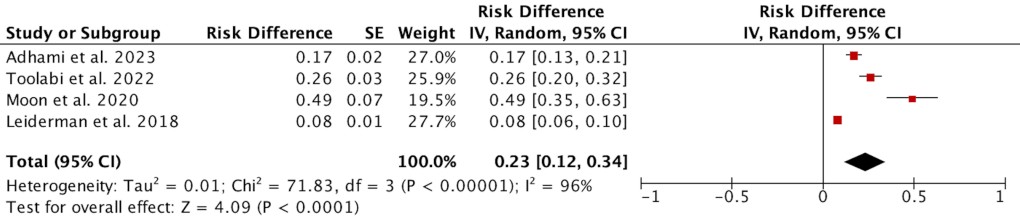

**Figure 4** Forest plots of severe compensatory hyperhidrosis rate after endoscopic thoracic sympathectomy. Note. *Adhami & Bell, 2023*; *Toolabi et al., 2022*; *Moon et al., 2020*; *Leiderman et al., 2018*.

[0.96–3.18], $p = 0.0003$; Fig. 3E). We performed a subgroup analysis to determine whether the studies targeted Asian populations, but we did not find a source of heterogeneity (Fig. S8). Besides, subgroup analyses based on varying definitions of CH (Fig. S9) and different follow-up durations (Fig. S10) did not reveal any significant BMI differences between the groups. Only two studies reported the season in which surgery was performed for a total of 268 patients. The random effects model showed no significant difference in the incidence of CH between surgeries conducted in spring and summer compared to those in autumn and winter ($P = 0.84$; Fig. 3F). Four studies recorded whether 2,019 patients had plantar hyperhidrosis before surgery, and the random effects model indicated that the occurrence of CH was not related to plantar hyperhidrosis ($P = 0.31$; Fig. 3G). Sensitivity analysis for the above comparisons revealed no sources of heterogeneity.

## Occurrence of severe compensatory hyperhidrosis

Four studies involving a total of 1,618 patients reported on the occurrence of severe CH. Due to high heterogeneity ($I^2 = 96\%$), a random effects model was used for the analysis. The results indicated that the incidence of SCH was 0.23 (95% CI [0.12–0.34]; Fig. 4). Sensitivity analysis did not reveal any sources of heterogeneity. Subgroup analysis based on the study's origin in Asia, varying definitions of SCH, and different follow-up durations revealed no significant differences between the subgroups (Fig. S11). We subsequently conducted a meta-regression considering the study's origin in Asia, different definitions of SCH, and follow-up periods, yielding $p$-values of 0.712, 0.937, and 0.803, respectively, indicating that these three factors are not sources of heterogeneity.

All four studies recorded the age of patients. Due to high heterogeneity, a random effects model was used for the analysis. The results indicated that there was no significant statistical difference in age between patients with and without SCH, with a WMD of 3.16 (95% CI [−0.39–6.72], $p = 0.08$; Fig. 5A). However, subgroup analysis revealed a notable difference in patient age between studies from Asia and those from non-Asia, while no significant age differences were observed among patients defined by varying definitions of SCH or different follow-up durations (Fig. S12). In terms of gender, there were 669 male patients and 949 female patients, but there was no statistical relationship between the occurrence of SCH and gender ($P = 0.11$; Fig. 5B). Two articles reported data on family history from 488 patients, with 266 having a family history and 222 not having one.

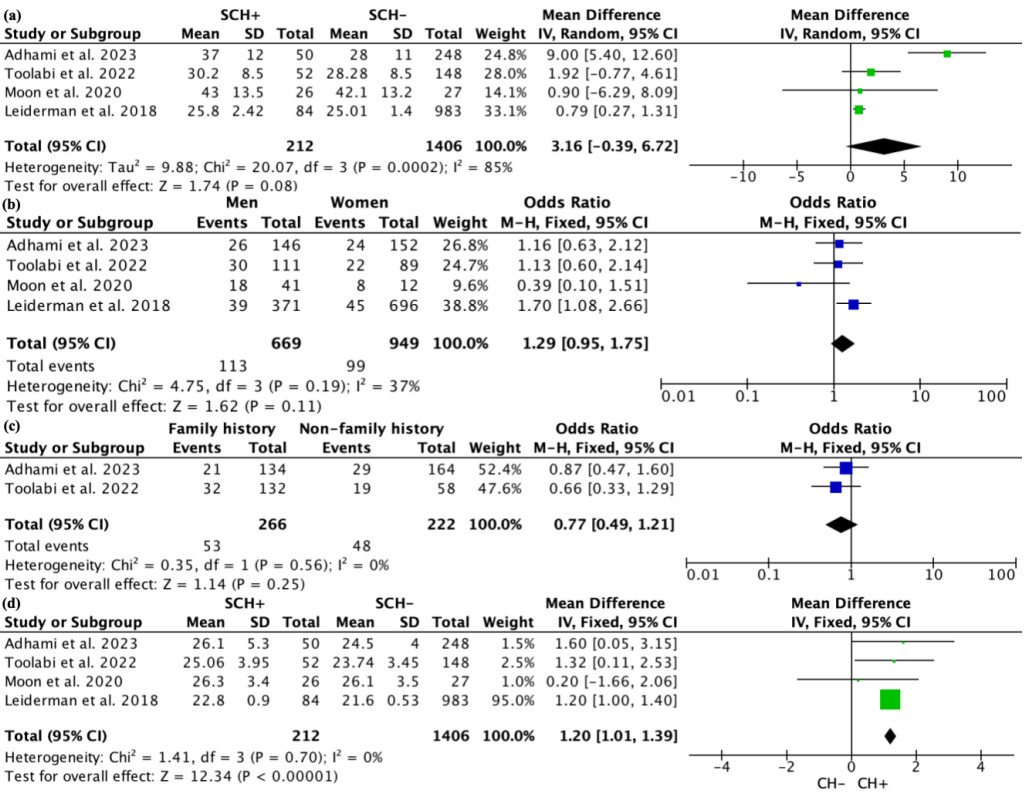

**Figure 5 Forest plot comparing clinical parameters between patients with and without severe compensatory hyperhidrosis after e ndoscopic thoracic sympathectomy.** Note. *Adhami & Bell, 2023*; *Toolabi et al., 2022*; *Moon et al., 2020*; *Leiderman et al., 2018*.

The fixed effects model showed no difference in the occurrence of CH based on family history ($P = 0.25$; Fig. 5C). Sensitivity analysis for the above comparisons did not reveal any sources of heterogeneity.

All four studies recorded the BMI of patients. Due to low heterogeneity ($I^2 = 0$), a fixed effects model was used, and it was found that patients with SCH had a higher BMI compared to those without SCH, with an SMD of 1.20 (95% CI [1.01–1.39], $p < 0.0001$; Fig. 5D). Subgroup analysis based on the study's origin in Asia, differing definitions of SCH, and varying follow-up periods found no significant differences in BMI between the subgroups (Fig. S13).

# DISCUSSION

This study employs a meta-analysis approach to evaluate the association between certain common clinical characteristics and the incidence of CH and SCH following ETS. To our knowledge, this is the first systematic review and meta-analysis assessing factors other than surgical techniques in relation to the incidence of CH and SCH after ETS. A total of 10 studies were included, examining eight relevant factors. We found that older patients and

those with a history of smoking had a higher incidence of CH, while patients with a higher BMI were more likely to experience both CH and SCH.

The precise mechanism underlying CH remains unclear, though it is widely thought to involve thermoregulatory responses and the reorganization of the autonomic nervous system (*Lin et al., 1998*) Following sympathetic nerve excision, the capacity for sweating in the affected skin area diminishes, prompting the remaining sympathetic nerves to functionally reorganize, which results in enhanced gland activity in other regions to offset heat dissipation (*Lin et al., 1998*; *Hsu et al., 1994*). With aging, physiological functions, including sweat gland activity and autonomic nervous system responsiveness, undergo changes that may affect the body's ability to regulate heat and, consequently, sweating patterns (*Inoue et al., 1999*). The most suitable patients for ETS are individuals who began experiencing hyperhidrosis symptoms prior to 16 and have the procedure performed before turning 25 (*Cerfolio et al., 2011*). Research by *Weksler et al. (2009)* identified age as a significant predictor for the development of CH post-ETS, noting a reduced incidence in younger individuals. Another study by *Bell, Jedynak & Bell (2014)* observed that the incidence of CH in patients under 18 years old post-ETS is less than 50%, while in those over 18, the incidence exceeds 80%. Moreover, advancing age may correlate with secondary causes of excessive sweating, including hypertension, diabetes, hyperthyroidism, and other endocrine issues (*Dufour & Candas, 2007*). Consequently, individuals over the age of 25 experiencing symptoms for fewer than five years should receive a formal assessment before admission to exclude possible systemic conditions leading to excessive sweating (*Miller et al., 2009*).

This research further demonstrates that patients with elevated BMI are at a higher risk for developing CH and SCH, a finding corroborated by multiple prior studies (*Alkosha et al., 2023*; *Leiderman et al., 2018*). Research by *Kargi (2017)* indicates that individuals with a BMI > 30 experience more severe sweating compared to the general population and are more likely to develop SCS. This may be due to the more active sweat gland activity in obese patients, as well as excessive adipose tissue potentially interfering with normal heat dissipation mechanisms, placing the body in a state of higher thermal load, thereby increasing the demand for compensatory sweating from other areas post-surgery to meet heat dissipation needs (*Miller et al., 2009*; *De Campos et al., 2005*). The STS consensus recommends that patients should keep their BMI below 28 prior to ETS surgery (*Cerfolio et al., 2011*). *Vannucci & Araújo (2019)* argue that the ideal ETS patient should have a BMI < 25 and strongly recommend that these patients maintain a normal weight even post-surgery to reduce the risk of postoperative CH. Therefore, an increase in BMI may be an important factor for the occurrence of CH, and besides informing patients with high BMI about the risk of CH before surgery, thoracic surgeons should also focus on controlling BMI preoperatively.

Individuals with a history of smoking are more likely to experience CH after ETS. *Kara et al. (2019)* found through multivariate regression analysis that smoking history is a significant predictor of CH, a conclusion that was reaffirmed in the study by *Masarwa et al. (2024)*. This may be attributed to nicotine in tobacco influencing the autonomic nervous system, increasing sympathetic nervous activity, which alters the body's sweating patterns

and subsequently raises the risk of CH (*Cramer & Jay, 2012*). Research by *Middlekauff, Park & Moheimani (2014)* proposed that nicotine and fine particulate matter in tobacco smoke can lead to an imbalance in the autonomic nervous system, favoring sympathetic activation, which becomes persistent through a positive feedback loop. Moreover, smoking generates reactive oxygen species (ROS), leading to oxidative stress, which may adversely affect sweat gland function and sweating patterns (*Kellogg, 2006*; *Pryor & Stone, 1993*).

Besides, several studies have indicated that other factors might have a potential impact on the incidence of CH following ETS. For example, research conducted by *Hyun et al. (2023)* and *Moon et al. (2020)* found that heart rate variability (HRV) serves as a non-invasive method for measuring autonomic nervous system activity, particularly the balance between sympathetic and parasympathetic branches. A reduction in HRV typically indicates heightened sympathetic activity, whereas an increase in HRV reflects augmented parasympathetic activity. Utilizing HRV assessments in conjunction with machine learning analyses demonstrates potential in predicting CH occurrence. *Carvalho et al. (2022)* highlighted that patients with axillary hyperhidrosis co-occurring with hyperhidrosis are more susceptible to developing CH in the thoracic and lumbar regions. Nevertheless, because of the restricted number of reports, we cannot incorporate them into our meta-analysis. In conclusion, while ETS has been broadly implemented in the treatment of hyperhidrosis, the incidence of postoperative CH continues to warrant attention, especially for patients with identifiable risk factors.

The non-surgical treatment of palmar hyperhidrosis has garnered considerable attention. Oxybutynin has been utilized effectively in cases of axillary and palmar hyperhidrosis, demonstrating commendable long-term efficacy (*Wolosker et al., 2014*). In contrast to surgical interventions, the use of Oxybutynin poses no risk of compensatory hyperhidrosis; however, numerous patients have reported experiencing dry mouth as a side effect. Nevertheless, Oxybutynin is generally associated with favorable outcomes and an enhancement in patient satisfaction. In addition to Oxybutynin, there are reports of sertraline being employed to address generalized social anxiety disorder accompanied by blushing complaints (*Jadresic et al., 2011*). Prior to surgical intervention, it may be prudent to consider a trial of pharmacotherapy for patients, as such treatment could potentially mitigate the risk of postoperative regret for both patients and physicians.

To our knowledge, this is the first systematic review and meta-analysis aimed at evaluating the association between factors other than surgical techniques and the incidence of CH and SCH after ETS. Our meta-analysis inevitably has certain limitations. Firstly, common surgical methods used in ETS include cauterization and the use of metal clips (*Inan et al., 2008*). Cautery, for example, typically results in more extensive nerve destruction, which could potentially lead to more significant compensatory sweating in other areas post-surgery. On the other hand, the use of metallic clips might offer a more localized effect on the sympathetic chain, which could result in a different pattern of postoperative sweating. Our study focuses solely on the most prevalent method, cauterization of the sympathetic chain, without extensively addressing the impact of other surgical methods on the occurrence of CH and SCH. This may overlook the potential influence of differences in surgical techniques on the results. Future studies exploring different techniques and

their respective impacts on CH and SCH could provide a more holistic understanding of the relationship between surgical methods and postoperative complications. Secondly, the number of studies included in this meta-analysis was relatively small, limiting the generalizability of our findings. Finally, many of the included studies were retrospective in nature and may have failed to account for all potential confounding factors, such as the duration of hyperhidrosis symptoms before surgery or pre-existing comorbidities, which could influence the outcomes.

In summary, our research offers robust evidence about the clinical traits of patients at increased risk for CH and SCH following ETS for hyperhidrosis. This assists thoracic surgeons in performing more comprehensive preoperative assessments for patients and provide better preoperative guidance who need ETS. It helps patients in accurately recognizing the potential risks of CH or SCH associated with ETS, thereby minimizing disappointment with postoperative outcomes. For older patients or those with a history of smoking, it is essential to inform them of the increased likelihood of CH before surgery and recommend a smoking cessation plan to minimize the impact of nicotine on the autonomic nervous system. Patients with a high BMI not only face an increased risk of CH but may also be at higher risk for SCH. It is recommended that they lose weight before surgery to maintain a lower BMI.

## ACKNOWLEDGEMENTS

We gratefully acknowledge Mr. Hong Xiao from the Department of Andrology and Sexual Medicine at the First Affiliated Hospital of Fujian Medical University, Fuzhou, China, for his substantial contributions to the literature review, data analysis, and filtering in this study.

### Funding

The authors received no funding for this work.

### Competing Interests

The authors declare there are no competing interests.

### Author Contributions

- Zhi-yi Lin conceived and designed the experiments, performed the experiments, analyzed the data, prepared figures and/or tables, authored or reviewed drafts of the article, and approved the final draft.
- Min Lin conceived and designed the experiments, performed the experiments, analyzed the data, prepared figures and/or tables, authored or reviewed drafts of the article, and approved the final draft.

### Data Availability

This is a systematic review/meta-analysis.

## Supplemental Information

Supplemental information for this article can be found online at http://dx.doi.org/10.7717/peerj.19097#supplemental-information.

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
