# Peer review of "Which patients are more likely to experience compensatory hyperhidrosis after endoscopic thoracic sympathectomy: a meta-analysis and systematic review"

_PeerJ, doi:10.7717/peerj.19097_

## Round 0.1 · original submission · Minor Revisions

Kindly address all the comments from the reviewers in the revised manuscript and provide a point-by-point response in a separate file.

Reviewer 1 ·

Basic reporting

The manuscript presents a meta-analysis and systematic review focusing on compensatory hyperhidrosis (CH) following endoscopic thoracic sympathectomy (ETS). The study is timely and addresses a significant clinical issue, providing valuable insights into patient factors that may predict the development of CH and severe compensatory hyperhidrosis (SCH). The methodology is robust, and the manuscript is generally well-structured. However, there are areas for improvement in terms of the manuscript.
1.The language is generally clear, but there are occasional grammatical errors and awkward phrasings. For example, in the introduction, the phrase "potentially influenced by genetic factors, climate conditions, and lifestyle choices" could be rephrased for better flow.
2. The introduction effectively sets the stage for the study but could benefit from a more detailed discussion on the implications of CH and SCH on patient quality of life and the current gaps in knowledge.
3. The methodology is well-described, following PRISMA guidelines. However, the search strategy could be detailed in the main text rather than being relegated to supplementary material to enhance transparency.
4. The discussion is comprehensive, but the limitations of the study should be discussed in more detail.
5. The random effects model was appropriately used given the heterogeneity across studies. However, the authors should provide more detail on how heterogeneity was assessed and justify the use of specific statistical tests.
6. The use of Egger's test is appropriate, but the authors should discuss the implications of publication bias, especially for the SCH analysis, and consider sensitivity analyses or meta-regression to explore this further.
The manuscript is a valuable contribution to the field of thoracic surgery and hyperhidrosis research. With attention to the language, structure, and detailed presentation of methods and results, it has the potential to be a high-impact publication. I recommend a minor revision before reconsideration for publication.

Experimental design

no comment

Validity of the findings

no comment

Additional comments

Incorporate comprehensive quality of life assessments in future studies to better understand the impact of CH on daily functioning and to guide the development of patient-centered treatment strategies. Future research can contribute to a more comprehensive understanding of CH, leading to improved treatment strategies and better outcomes for patients undergoing ETS.

·

Basic reporting

NO COMMENT

Experimental design

NO COMMENT

Validity of the findings

NO COMMENT

Additional comments

GOOD MANUSCRİPT

DEAR AUTHOR/AUTHORS
YOUR ARTİCLE İS GENERALLY GOOD. İ THİNK YOU MUST MENTİON THE OPERATİVE TYPES İN DİSCUSSİON SECTİON (FOR EXAMPLE CAUTERY OR METALLİC CLİP)

KİNDLY REGARDS

Reviewer 3 ·

Basic reporting

This manuscript addresses a clinically significant issue: compensatory hyperhidrosis (CH) and severe compensatory hyperhidrosis (SCH) following endoscopic thoracic sympathectomy (ETS). The text is written well, arguments are presented concisely, without unnecessary complexity or redundancy. Related works are cited properly.

Experimental design

The manuscript follows Prospective Register of Systematic Reviews and PRISMA, meaning that it adheres to rigorous systematic review guidelines and minimizes bias in study selection while ensuring the meta-analysis process is transparent and replicable.

Validity of the findings

The use of statistical techniques, such as random-effects models, is appropriate for addressing the significant heterogeneity across studies. The findings are supported by rigorous statistical procedures and are therefore sound.

Reviewer 4 ·

Basic reporting

The article provides a comprehensive meta-analysis of compensatory hyperhidrosis (CH) and severe compensatory hyperhidrosis (SCH) following endoscopic thoracic sympathectomy (ETS). With data from 10 studies and 2,566 patients, the analysis presents critical insights into the incidence and contributing factors of CH and SCH. No comment

Experimental design

No comment

Validity of the findings

No comment

Additional comments

Points for Improvement

Limited Discussion on Variability Among Studies: While the meta-analysis includes 10 studies, the article could provide a more detailed discussion on heterogeneity, such as differences in patient populations, surgical techniques, or definitions of CH and SCH across studies.

Narrative Description Format: The reliance on narrative descriptions where pooling was not feasible might limit the ability to draw robust conclusions in these areas. Employing subgroup analysis could enhance these findings.

Incomplete Data Reporting: The abstract mentions that the total sample size is 3,117 patients, but the results focus on 2,566 patients. Clarifying the discrepancy would strengthen the study’s transparency.

Intervention Recommendations: While the study identifies risk factors, it lacks a discussion on how these findings can translate into clinical interventions or preventive strategies to mitigate CH and SCH.

Broader Context: Incorporating comparisons with alternative surgical approaches or non-surgical treatments for hyperhidrosis could provide a more holistic perspective.

Detailed Methodology: Providing a clearer description of how studies were selected, including inclusion and exclusion criteria, would bolster the reproducibility of the meta-analysis.

This article offers valuable insights into CH and SCH following ETS and identifies key risk factors associated with these conditions. However, addressing variability among studies, clarifying data discrepancies, and providing more actionable recommendations could significantly enhance the article's impact. Future research should focus on standardizing definitions of CH and SCH, exploring preventive measures, and comparing the outcomes of ETS with other treatments for hyperhidrosis.

·

Basic reporting

.

Experimental design

.

Validity of the findings

.

Additional comments

Dear author(s)
I thoroughly reviewed the paper titled "Which patients are more likely to experience compensatory hyperhidrosis after endoscopic thoracic sympathectomy: A meta-analysis and systematic review." I have some comments and suggestions.
1)The title "Which patients are more likely to experience compensatory hyperhidrosis after endoscopic thoracic sympathectomy: A meta-analysis and systematic review" is clear and descriptive. It identifies the main research focus and includes critical terms such as "compensatory hyperhidrosis," "endoscopic thoracic sympathectomy," and the study type ("meta-analysis" and "systematic review"). It effectively reflects the focus of the research. While specific, the title could emphasize key outcomes (e.g., "demographic and lifestyle factors") for added clarity.
2)The abstract summarizes the study's background, objectives, methodology, results, and conclusions concisely. It includes details on the study population, primary findings, and the implications of the result. The primary and secondary outcomes are clearly stated, and the significance of findings is emphasized. But please add the study period or database search timeframe for transparency and briefly mention the methodological rigor, such as adherence to PRISMA guidelines.
3) The introduction provides a strong foundation, describing the significance of compensatory hyperhidrosis and the role of endoscopic thoracic sympathectomy. It discusses gaps in current knowledge and introduces the study's objective. It is clear and contextualized with relevant literatüre and identifies the clinical importance and the potential impact of the research. However please cite more recent studies to strengthen the discussion of gaps in knowledge and clearly articulate how this meta-analysis differs from or builds upon previous work.
4)The study objectives are focused on identifying factors influencing compensatory hyperhidrosis following surgery.The objective aligns well with the research question. It is specific and measurable. Please condense the objective into a single sentence for clarity and emphasis.
5)The methods section adheres to systematic review and meta-analysis standards, following PRISMA guidelines and registering a protocol with PROSPERO. It details inclusion criteria, search strategy, data extraction, quality assessment, and statistical analysis. The methodology is comprehensive and reproducible. Statistical analyses are well-described and appropriate for the study design. But you could provide a rationale for excluding studies published in non-English languages. Please include a flowchart summarizing the selection process for ease of understanding.
6)The results are presented in a structured manner, highlighting the incidence of compensatory hyperhidrosis and related factors. Figures and forest plots are used to support quantitative findings. They are supported by statistical analysis, subgroup analyses, and sensitivity tests. The use of visuals like forest plots aids comprehension. But please enhance clarity by providing a summary table of key findings alongside statistical results and cearly separate results for primary and secondary outcomes to improve readability.
7)The discussion interprets findings in the context of existing literature, exploring potential mechanisms and implications. It addresses study limitations and clinical recommendations. It is well-balanced, acknowledging limitations like publication bias and identifies practical applications for thoracic surgeons.However You could expand on the discussion of age-related factors and their clinical implications and propose areas for future research based on identified gaps.
8)The conclusions succinctly summarize the findings, emphasizing key risk factors for compensatory hyperhidrosis and severe compensatory hyperhidrosis. This section is concise and actionable. The research question is answered, and the implications are clear. But please highlight the broader impact of the findings on surgical practice and patient counseling.
9) Please clarify and highlight methodology details, such as exclusion criteria rationale and PRISMA adherence, enhance result visualization with summary tables and expand the discussion on clinical and research implications.
Thank you.

---

## Round 0.2 · accepted · Accept

Authors have addressed all of the reviewers' comments and manuscript is ready for publication.

·

Basic reporting

NO COMMENT

Experimental design

NO COMMENT

Validity of the findings

NO COMMENT

Reviewer 4 ·

Basic reporting

The necessary changes have been made as per the initial review. The reporting is clear and without ambiguity. English language and flow was improved.

Experimental design

The methods part of the manuscript are precise and reproducible, as it is listed in the figure.

Validity of the findings

The research is very robust with improved references. The initial review indications were met and the underlying data was provided. The conclusions part is well stated and underlines the study.

Additional comments

-

·

Basic reporting

The revised article can be accepted.

Experimental design

The revised article can be accepted.

Validity of the findings

The revised article can be accepted.

Additional comments

The revised article can be accepted.